# Effect of Interface Coating on High Temperature Mechanical Properties of SiC–SiC Composite Using Domestic Hi–Nicalon Type SiC Fibers

**Enze Jin [1], Wenting Sun [1], Hongrui Liu [1,2], Kun Wu [1], Denghao Ma [1], Xin Sun [1], Zhihai Feng [1], Junping Li [1,\*] and Zeshuai Yuan [1]**

[1] Key Laboratory of Advanced Functional Composites Technology, Aerospace Research Institute of Materials & Processing Technology, Beijing 100076, China; jinenzelc@163.com (E.J.); wentingsun2008@163.com (W.S.); liu_hongrui@126.com (H.L.); wukunjez@163.com (K.W.); madenghao1987@163.com (D.M.); sunxinhit@163.com (X.S.); fengzhihaijez@163.com (Z.F.); fnl37051123@163.com (Z.Y.)

[2] Science and Technology on Advanced Ceramic Fibers and Composites Laboratory, College of Aerospace Science and Engineering, National University of Defense Technology, Changsha 410073, China

\* Correspondence: jpli@iccas.ac.cn

**Abstract:** Here we show that when the temperature exceeded 1200 °C, the tensile strength drops sharply with change of fracture mode from fiber pull-out to fiber-break. Theoretical analysis indicates that the reduction of tensile strength and change of fracture mode is due to the variation of residual radial stress on the fiber–matrix interface coating. When the temperature exceeds the preparation temperature of the composites, the residual radial stress on the fiber–matrix interface coating changes from tensile to compressive, leading to the increase of the interface strength with increasing temperature. The fracture behavior of SiC–SiC composites changes from ductile to brittle when the strength of fiber–matrix interface coating exceeds the critical value. Theoretical analysis predicts that the high temperature tensile strength can increase with a decrease in fiber–matrix interface thickness, which is verified by experiments.

**Keywords:** ceramic matrix composite; high temperature strength; SiC fiber; fiber–matrix interface coatings; residual stress

## 1. Introduction

Continuous SiC fiber-reinforced SiC matrix composite (SiC–SiC) is known as a high-temperature resistant material, with excellent properties such as high specific strength, high specific stiffness, high temperature resistance, long-term oxidation resistance and erosion resistance. Therefore, SiC–SiC composite has a wide application prospect in the thermal protection system of aerospace vehicles and hot-end components of aircraft engines [1–3]. In addition, SiC–SiC composite with near stoichiometric (3rd generation) SiC fibers has high irradiation resistance, which is one of the most promising materials for structural application in nuclear fusion reactor and nuclear fission power system [4–6].

In the view of fracture mechanics of composites, fiber–matrix interface is the most important factor that determines the structural stability of a composite. Recently, intensive efforts have been devoted to study the influences of interface on mechanical properties of SiC–SiC composites. Wang et al. studied the tensile creep properties of 2D-SiC–SiC composites in vacuum at high temperature [7]. The results show that the interface strength is relatively strong and the matrix cracking is inhibited at 1300 and 1350 °C. The roles of fiber on the creep behaviors of composites are determined by the fiber–matrix interface. Buet et al. systematically analyzed the fiber–matrix interface of SiC–SiC reinforced by stoichiometric SiC fibers. Their experiments demonstrate that the interface shear stress

depends on the texture of the carbon interface. Highly anisotropic pyrocarbon interfaces increase the interfacial shear stress [8]. They also found that Tyranno SA3 fiber has a granular and rough surface which leads to an increase of the interface shear strength. The Tyranno SA3-based composites with stronger interface exhibit a brittle behavior [9]. Fellah et al. investigated the influence of the carbon interface on the mechanical behavior of SiC–SiC [10]. The results show that the fiber–matrix debonding behavior depends strongly on the nature of the carbon on the SiC fiber surface, which is different according to the SiC fiber. Hsu et al. measured the interface tensile strength of SiC/Si directly for the first time by taking advantage of the FIB method [11].

It is well acknowledged that appropriate interface can effectively transfer load and inhibit crack propagation. Ding et al. studied the influence of PIP–SiC interface on the mechanical properties of SiC–SiC composites at high temperature [12]. The results show that the PIP–SiC interface can improve the toughness of the composite by deflecting most cracks in matrix. Wang et al. developed SiC–SiC composites with a PyC–SiC multilayer interface [13]. The results reveal that when the composites are oxidized at the temperature higher than 900 °C, the composites exhibit self-healing characteristics. Shimoda et al. developed non-brittle fractures in SiC–SiC composites without a fiber–matrix interface [14]. They found that sandwiched layers with a porous matrix in a laminate can achieve crack deflection and exhibit ductile fracture behavior. Therefore, in order to provide a deep insight on the stability of SiC–SiC composite, it is important to investigate the effect of interface coating on the mechanical properties.

Domestic Hi–Nicalon type SiC fiber is a new material and there are few studies on the its composite. In this study, we investigate the tensile properties of SiC–SiC composites at high temperature reinforced by domestic Hi–Nicalon type SiC fiber and illuminate effect of interface coating on the mechanical property variations by combining theoretical analysis and experiments.

## 2. Materials and Methods

### 2.1. Preparation of SiC–SiC Composites

Two-dimensional preform was woven using domestic Hi–Nicalon type SiC fibers. The SiC fibers were provided by Xiamen University (Xiamen, China). The fiber diameter is 14 µm, density is 2.79 g/cm, tensile strength is 2.7 GPa, and the modulus is 270 GPa. The typical chemical composition of the fiber is listed in Table 1. The properties of the domestic Hi–Nicalon type SiC fiber are close to those of Hi–Nicalon fiber [15,16]. Pyrolytic carbon (PyC) layers with a thickness of about 580 nm were coated on the surface of the preforms by chemical vapor deposition (CVD) method. SiC–SiC composites were prepared by precursor infiltration and pyrolysis (PIP) process. The preforms were impregnated with liquid state Polycarbosilane (PCS) by a vacuum infiltration method and pyrolyzed at 850 °C in an inert Argon atmosphere. The impregnation and pyrolysis process were repeated 10 times until weight increase was less than 1%. The final porosity of the composites is 6%–9%.

**Table 1.** Typical chemical composition of domestic Hi–Nicalon type SiC fibers.

| Si Content Wt % | C Content Wt % | O Content Wt % | C/Si Mole Ratio |
|---|---|---|---|
| 61.5 | 37.9 | 0.6 | 1.44 |

### 2.2. Tensile Test of SiC–SiC Composites at High Temperature in Air

The samples used in the high temperature tensile test are shown in Figure 1. At the clamping ends, two reinforcing pieces were stuck to both sides of the specimen in case of clamping failure during the experiment. Tensile tests were performed in the Instron 1332 equipment. A four-zone controlled high-temperature furnace was used. The temperature distribution on the gauge was well controlled within ±5 °C. The loading rate of tensile test was 2 mm/min. The heating rate was 30 °C /min and the holding time was 20 min. Five specimens were tested for each state and the strength was obtained by averaging the values over these five results.

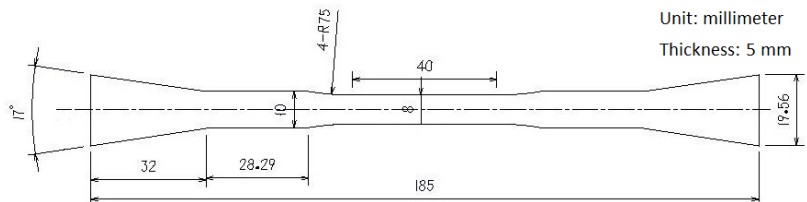

**Figure 1.** Sketch of the tensile test specimen.

## 3. Results and Discussion

### 3.1. Tensile Strength of SiC–SiC Composites at High Temperature in Air

The tensile strength of SiC–SiC composite was tested in situ at room temperature (RT), 800, 1200, 1300 and 1500 °C, respectively. The stress–strain curves at different temperatures are shown in Figure 2a. Each stress–strain curve is taken from the results with closet values to the average of all the five specimens at each temperature. The variations of tensile strength with temperature are summarized in Figure 2b. It can be seen from the results that the stress–strain curve barely changes at 800 °C. However, when the temperature further increases to 1200, 1300 and 1500 °C, the tensile strength decreases significantly by 48%, 58% and 66%, respectively. The tensile modulus slightly decreases with temperature from RT to 1300 °C. There is an obvious drop of tensile modulus at 1500 °C. The tensile modulus of composite is mainly determined by the modulus of fibers according to the composite material mechanics. When the temperature exceeds the preparation temperature of domestic Hi–Nicalon type SiC fibers (~1350 °C), the fiber modulus begins to degrade, leading to the obvious drop of SiC–SiC tensile modulus at 1500 °C.

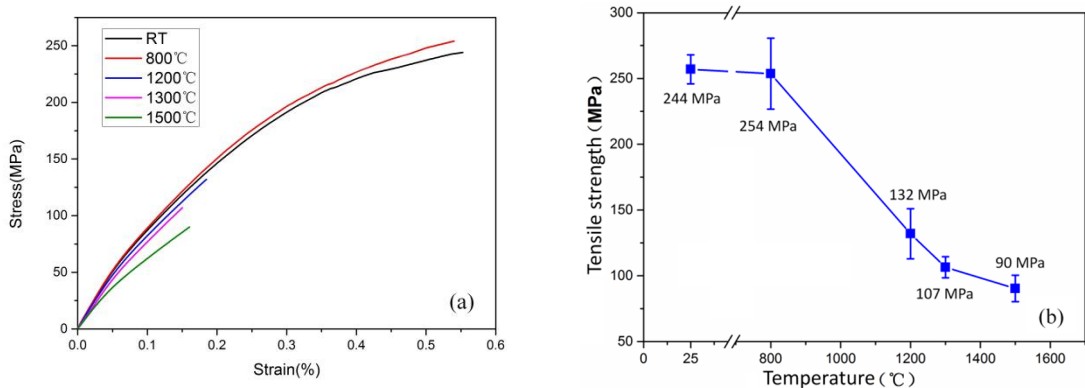

**Figure 2.** (**a**) Tensile stress–strain curves and (**b**) variations of tensile strength of SiC–SiC at different temperatures in air.

The microstructures at the fracture of SiC–SiC composites were characterized by Camscan Apollo 300 scanning electron microscope (CamScan, Cambridge, UK). The fractures of SiC–SiC composite at 800 °C exhibit ductile characteristics as shown in Figure 3. A large number of pulled out fibers with length of several millimeters can be seen at the fracture. When the temperature reaches 1200 °C, the length of pulling out fiber decreases to about tens of microns. When the temperature further increases to 1300 °C, the length of pulling out fibers further decreases and the regions of pulling out fibers become smaller. Most of the fibers are not pulled out, but break along with the matrix. When the temperature reaches 1500 °C, the fractures show typical brittle characteristics. The fracture surface of the composites is very plane and no pullout fibers can be seen. Cracks are not arrested at the interface, but penetrate the fiber bundles at 1500 °C.

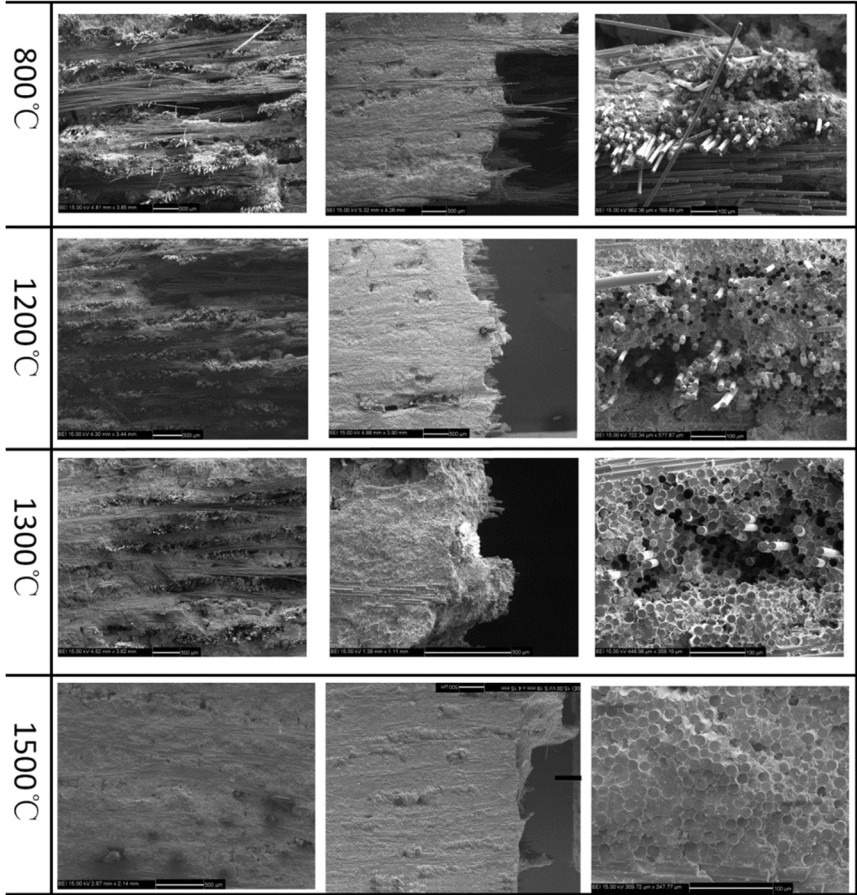

**Figure 3.** Tensile fracture morphology of SiC–SiC composites at high temperature in air.

### 3.2. Tensile Strength of SiC Fiber at High Temperature

In order to investigate the decline of tensile strength of SiC–SiC composites at high temperature, the tensile strength of monofilament SiC fiber at 1300 °C in air was tested in situ on the fiber ultra-high temperature performance testing system in Aerospace Research Institute of Materials & Processing Technology (Beijing, China). The gauge length was 40 mm. Monofilament SiC fiber was heated to 1300 °C with a heating rate of 40 °C/min and annealed for 5 min. The loading rate of tensile testing was 4 μm/s. The results are shown in Table 2. It can be seen that SiC fibers still retain high strength at 1300 °C and the strength retention rate is 78.9%, which is close to that of Hi–Nicalon fibers [17]. It indicates that the degradation of fiber is not the key factor that causes the sharp reduction of tensile strength of SiC–SiC composites at high temperature.

**Table 2.** Tensile strength of the SiC fiber at high temperature.

| Room Temperature | | 1300 °C in Air | | Strength Retention Rate (%) |
|---|---|---|---|---|
| Strength (GPa) | Cv Value (%) | Strength (GPa) | Cv Value (%) | |
| 2.8 | 11.9 | 2.21 | 23.6 | 78.9 |

### 3.3. Influence of Fiber–Matrix Interface Coating on Tensile Properties of SiC–SiC Composites

The interface is known to play an important role in mechanical properties of materials [8,10,18,19]. According to the theory of fracture mechanics, the existence of fiber in the composite inhibits the crack propagation and improves the toughness of the material. In this study, it is assumed that both SiC fiber and SiC matrix is linear elasticity, and the relationship between interface shear stress and shear displacement is also linear. The contribution of fibers to the fracture toughness of composites is

expressed as $\Delta K$. Based on the fracture mechanics theory, the relationship between $\Delta K$ and interface strength $\tau_b$ is shown in formula (1) [20]

$$\Delta K = \begin{cases} \sqrt{\dfrac{d(E_mA_m+E_fA_f)\tau_b}{2(E_mA_mE_fA_f)}} \dfrac{E_fA_f\delta_b^{3/2}}{(A_f+A_m)\eta K_{IC}} \dfrac{\alpha}{\sqrt{1-\alpha^2}} (\pi - 2arcsin\alpha) & \left(\tau_b < \tau_b^C, \text{ pull out}\right) \\ \sqrt{\dfrac{2(E_mA_mE_fA_f)\delta_b}{d(E_mA_m+E_fA_f)\tau_b}} \dfrac{(\sigma_f^b)A_f}{(A_f+A_m)\eta K_{IC}E_f} & \left(\tau_b > \tau_b^C, \text{ break}\right) \end{cases} \tag{1}$$

where, $d$ is the diameter of the fiber, $E_m$ and $E_f$ are the Young's modulus of the matrix and fiber, $A_m$ and $A_f$ are the equivalent cross section areas of the matrix and fiber: $A_f = \pi d^2$, $A_m = d^2 - \pi d^2$, $\sigma_f^b$ is the tensile strength of the fiber, $\tau_b$ is the interface strength, $\eta = 2\sqrt{2}(1 - v_m^2)/(E_m\sqrt{\pi})$, where $v_m$ is the Poisson's ratio of the matrix [21], $\delta_b$ is the critical interface shear displacement [22], $K_{IC}$ and is the fracture toughness of the matrix. $\alpha = E_mA_m/(E_fA_f)$. $\tau_b^C$ Is the critical interface strength of fiber pull-out/break transformation, as shown in formula (2):

$$\tau_b^C = \frac{(\sigma_f^b A_f)^2}{CE_m^2 A_m^2 \delta_b} \tag{2}$$

$$C = \pi d \left(\frac{1}{E_fA_f} + \frac{1}{E_mA_m}\right) \tag{3}$$

The parameters involved in the formula above are shown in Table 3. We investigated the mechanical properties of SiC fiber and SiC matrix by nano-indentation tests. The results reveal that the modulus of SiC matrix is close to that of SiC fiber. For this reason, in this study, we assume that the SiC matrix has the same mechanical properties as SiC fiber for simplicity.

**Table 3.** Physical parameters used in fracture toughness calculation [21,22].

| $d$ μm | $E_f$ GPa | $E_m$ GPa | $v_m$ - | $\sigma_f$ GPa | $\delta_b$ μm |
|---|---|---|---|---|---|
| 14 | 270 | 270 | 0.15 | 3 | 2 |

By substituting the data in Table 3 into Formulas (1) and (2), the relation of fracture toughness enhancement $\Delta K$ as a function of fiber–matrix interface strength can be obtained, as shown in Figure 4.

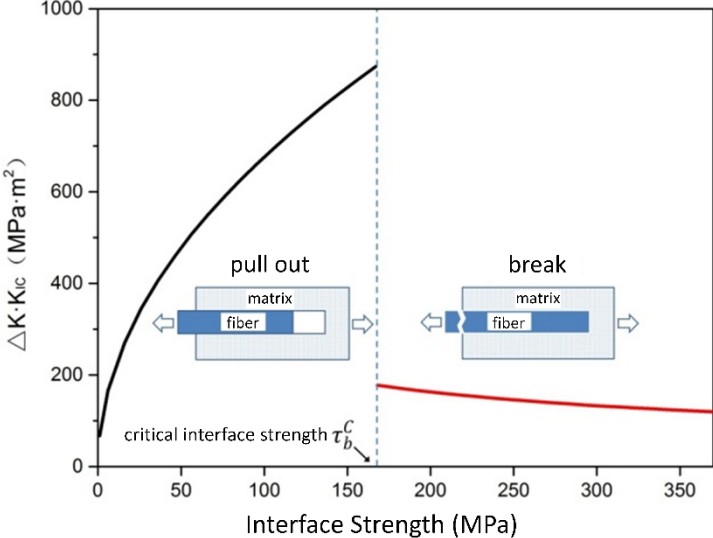

**Figure 4.** Fracture toughness enhancement of SiC–SiC as a function of fiber–matrix interface strength.

It can be seen from Figure 4 that when the failure mode is fiber pull-out, the toughening effect of the fiber increases with an increase in interface strength. Once the interface strength exceeds the critical value of fiber pull-out/break transition, the fracture toughness of SiC–SiC will drop suddenly. With the fiber-break failure mode, the fracture toughness of SiC–SiC decreases gradually with an increase in interface strength. For this reason, it is necessary to avoid the fiber-break failure mode for SiC–SiC composites. If SiC–SiC composites break in the fiber pull-out failure mode, further enhancement of the interface strength will improve the toughness of the material. Conversely, if SiC–SiC composites break in the fiber-break failure mode, the reduction of interface strength will improve the toughness of the material.

In SiC–SiC composites, the thermal expansion coefficient of the PyC interface coating is different from those of SiC fiber and SiC matrix. Therefore, the variation of temperature during PIP process will cause residual stress in the composites. Based on the characteristics of SiC–SiC composites, a two-dimensional finite element model of SiC–SiC unit cell is established, as shown in Figure 5. The model is composed of SiC fiber, PyC interface coating and SiC matrix. The fiber volume fraction is 40%. The material parameters used in the calculation are shown in Table 4. It is assumed that SiC fiber and SiC matrix are isotropic materials. The PyC interface coating is set to be transversal isotropy. The Young's modulus, sheer modulus, Poisson's ratio and coefficient of thermal expansion of PyC are determined according to the reference [23]. The Young's modulus of fiber is determined by our test. The Poisson's ratio of fiber is determined according to the reference [21]. The coefficient of thermal expansion of SiC fiber is determined by our test. We assume that the SiC matrix have the same mechanical properties as the SiC fiber. The densification process of PIP SiC matrix during heating above preparation temperature (850 °C) is not considered in this calculation. The thickness of the interface coating is 850 nm. The elements used are four-node bilinear plane stress elements (CPS4R), and the calculation is performed by Abaqus software. The simulation is divided into three analysis steps: 1. Set the initial analysis step. The initial temperature is set to 850 °C (SiC–SiC composite preparation temperature in this study). 2. Simulate the cooling process of SiC–SiC composites to room temperature (temperature field decreases from 850 to 25 °C). 3. Simulate the heating process tensile test of SiC–SiC composite to different temperatures and calculate the residual stress.

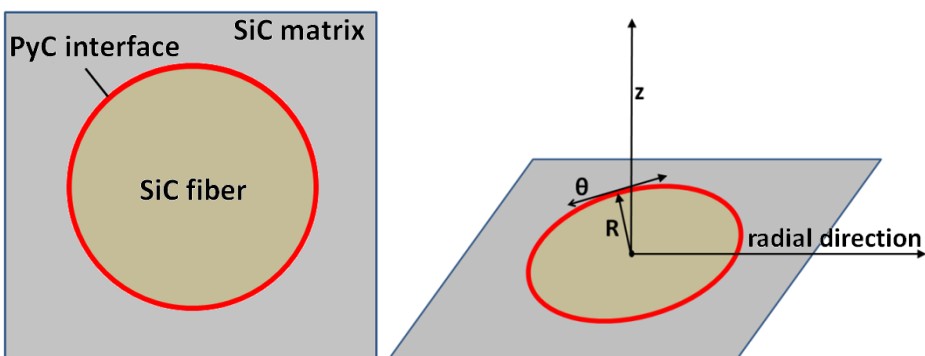

**Figure 5.** Sketch of SiC–SiC FEM model.

**Table 4.** Mechanical parameters used in the finite element calculation of residual stress [21,23].

| Material | Young's Modulus (GPa) | | Poisson's Ratio | | Coefficient of Thermal Expansion ($10^{-6}$ °C$^{-1}$) | |
|---|---|---|---|---|---|---|
| | $E_{zz}$ | $E_{RR}$ | $V_{R\theta}$ | $V_{Rz}$ | $\alpha_{zz}$ | $\alpha_{RR}$ |
| SiC fiber | 270 | 270 | 0.15 | 0.15 | 3.5 | 3.5 |
| PyC interface | 30 | 12 | 0.12 | 0.4 | 2 | 28 |
| SiC matrix | 270 | 270 | 0.15 | 0.15 | 3.5 | 3.5 |

During the tensile process, crack initiations in SiC–SiC composites mainly appear in the matrix and gradually propagate to the fiber–matrix interface coating. Experiments revealed that the failure mode of SiC–SiC composites is controlled by the interface strength [10]. If the fiber–matrix interface coating is weak enough, the crack will deflect into the interface and propagate along fiber axis direction. If the interface coating is too strong, the crack will not deflect, but directly propagate through the fibers. According to the Mohr–Coulomb criterion, shear strength is negatively correlated with the compressive stress perpendicular to the shear plane [24]. Therefore, the residual radial stress at the interface, $\sigma_{RR}$, has an important effect on the interface strength. The distribution of residual radial stress in SiC–SiC composites is shown in Figure 6.

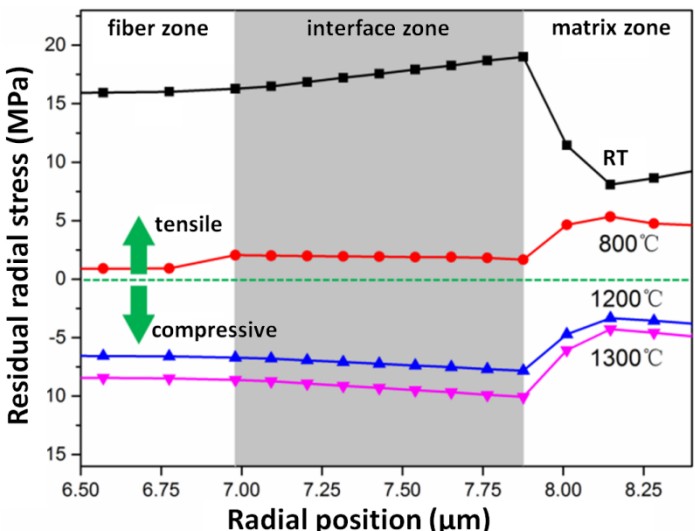

**Figure 6.** Distribution of residual radial stress,$\sigma_{RR}$ of SiC–SiC at different temperature.

It can be seen clearly that the residual radial stress decreases with temperature. When the temperature exceeds 850 °C, the residual radial stress transforms from tensile to compressive, which prevents the crack propagation into the interface coatings. Therefore, at low temperature, the interface of SiC–SiC composite is relatively weak. Cracks tend to be deflected at the interface coatings, leading to fiber pull-out failure mode. With increase of temperature, the residual radial stress at the interface coatings decreases, which transforms from tensile to compressive. The interface becomes stronger at high temperature. Cracks can hardly be deflected and penetrate the fiber directly, leading to fiber-break failure mode and sudden reduction of tensile strength of SiC–SiC composite (as shown in Figure 7). Thus, one can attribute the transition from ductile to brittle at high temperature to the increase of interface strength.

The variations of residual hoop stress, $\sigma_{\theta\theta}$, with increasing temperature are shown in Figure 8. It can be seen from the results that below the preparation temperature, the residual hoop stress in the fiber and interface coatings is tensile. On the contrary, the residual hoop stress in the matrix is compressive. The residual hoop stress decreases with temperature. When the temperature exceeds preparation temperature, the residual hoop stress in the fiber and interface coatings converts to compressive and the residual hoop stress in the matrix converts to tensile. Above 850 °C, the residual hoop stress increases with temperature. According to the theory of fracture mechanics, compressive hoop stress in the interface coatings will restrain cracks extension. Therefore, the compressive hoop stress at high temperature will further enhance the interface and lead to tensile strength decline of the composites.

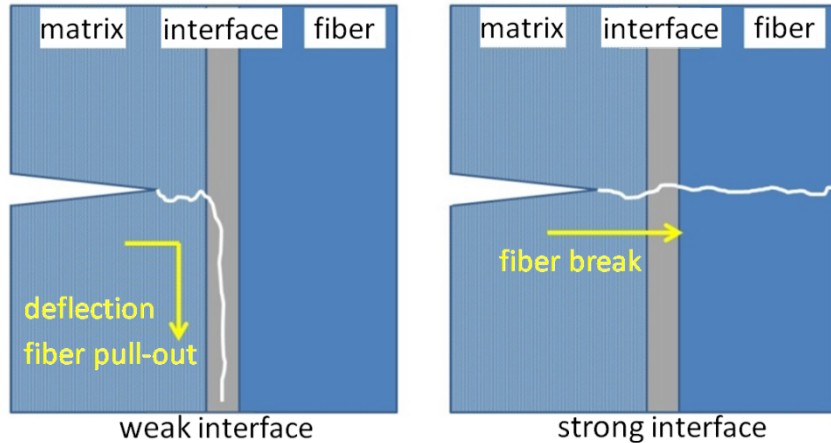

**Figure 7.** Schematic illustration of influence of interface strength on fracture mode of SiC–SiC.

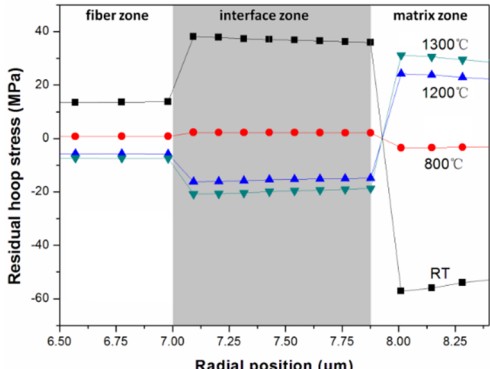

**Figure 8.** Distribution of residual hoop stress, $\sigma_{\theta\theta}$, of SiC–SiC at different temperature.

It is worthy to mention that when the temperature exceeds the preparation temperature (850 °C), the matrix will shrink due to PCS pyrolysis. This process will cause weight loss and crack initiations in the matrix, leading to acceleration of SiC–SiC strength reduction [10]. The chemical interactions between composite and air can also affect the mechanical properties of SiC–SiC at high temperature [11,12]. SiC matrix would form a glass phase at high temperature, which wraps the fiber and enhance the interface bonding [25]. The interface strength changes from weak to strong with increasing temperature by all factors discussed above and the composite changes from ductile fracture behavior to brittle fracture behavior.

We investigated the effect of interface coating thickness on the residual radial stress of SiC–SiC composites at 1300 °C. It can be seen from Figure 9 that at this temperature, the residual radial stress at the interface coating is compressive and decreases with an increase in interface thickness. The SiC–SiC composite breaks in fiber-break failure mode at 1300 °C. Therefore, reducing the thickness of the interface can reduce the interface strength and improve the fracture toughness of the SiC–SiC composites according to Figure 4.

To verify our theory, SiC–SiC composites with different interface thicknesses are prepared. The microstructures of SiC–SiC composites with average interface thickness of 100, 360, 580, and 640 nm are shown in Figure 10a. The tensile strength of SiC–SiC composites with different interface thicknesses at 1300 °C in air are shown in Figure 10b. It can be seen from the results that the tensile strength of SiC–SiC composite increases with decreasing interface thickness, in good agreement with the theoretical analysis above. It is necessary to mention that if there is no PyC interface coating, the SiC fiber and SiC matrix will sinter together at high preparation temperature. There will be strong interactions between fiber–matrix and the mechanical strength of the SiC-SiC composite is very low.

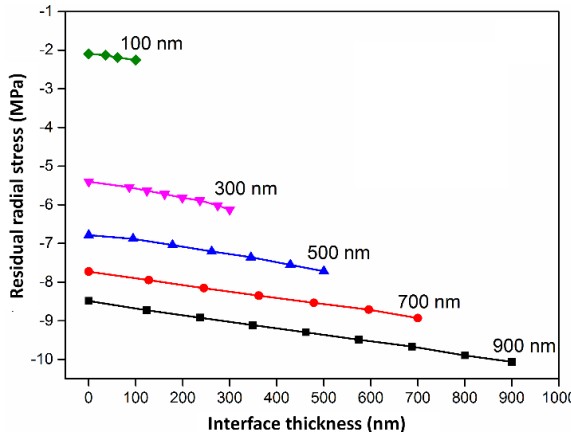

**Figure 9.** Distribution of residual stress,$\sigma_{RR}$ at 1300 °C with different interface thickness.

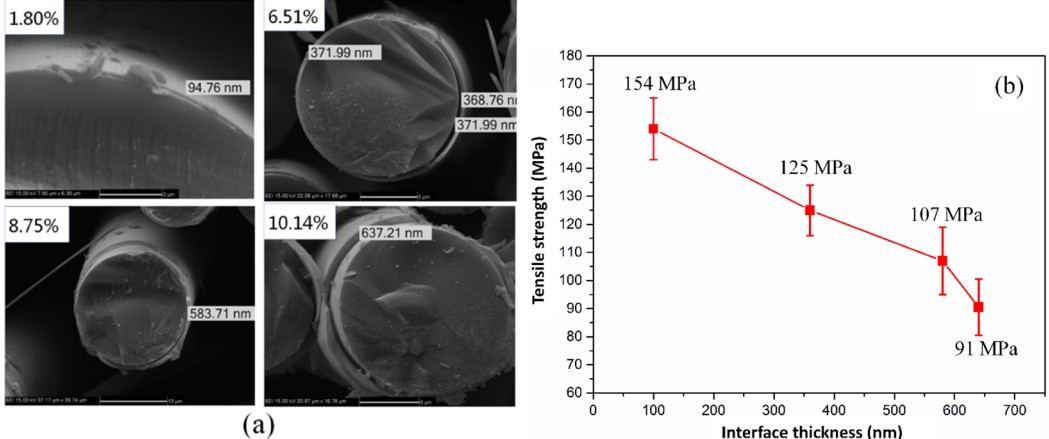

**Figure 10.** (**a**) SEM images of SiC–SiC composites with different interface thickness; (**b**) tensile strength of SiC–SiC composites with different interface thicknesses at 1300 °C in air.

## 4. Conclusions

- The tensile properties of SiC–SiC composites reinforced by domestic Hi–Nicalon type SiC fibers at 800, 1200, 1300, and 1500 °C were studied. The results show that when the temperature exceeded 1200 °C, the tensile strength of SiC–SiC composites decreased sharply and the composite failure mode converted from fiber-pull-out to fiber-break.
- Theoretical calculations showed that the toughening effect of fibers increases with interface strength when the composite failure mode is fiber-pull-out. Once the interface strength exceeds the critical value, the composite failure mode converts to fiber-break and the fracture toughness of SiC–SiC drops sharply. The finite element method simulations show that when the temperature exceeds the material preparation temperature, the residual radial stress at the interface increases and changes from tensile to compressive, causing transition of the failure mode and sudden reduction of tensile strength.
- Theoretical and experimental results showed that reducing the thickness of PyC interface coating improved the tensile strength of SiC–SiC composites at high temperatures.

**Author Contributions:** Conceptualization, E.J., X.S., Z.F., and J.L.; Data curation, W.S., H.L., D.M., and X.S.; Formal analysis, H.L., K.W., and J.L.; Funding acquisition, D.M. and Z.F.; Investigation, E.J., W.S., X.S., and J.L.; Methodology, E.J. and K.W.; Project administration, Z.F.; Resources, D.M.; Software, K.W.; Writing – original draft, E.J.; Writing – review & editing, W.S., H.L., X.S., J.L., and Z.Y. All authors have read and agreed to the published version of the manuscript.

**Funding:** This research received no external funding.

**Acknowledgments:** The authors thank Hongmei Yang, Lina Huang and Xinpeng Wang for outstanding work during experiments.

**Conflicts of Interest:** The authors declare no conflict of interest.

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
