# Peer review of "Effect of Interface Coating on High Temperature Mechanical Properties of SiC–SiC Composite Using Domestic Hi–Nicalon Type SiC Fibers"

_coatings, doi:10.3390/coatings10050477_

Round 1

Reviewer 1 Report

This article is of great interest but many modifications have to be applied for a rigorous analysis.

Analysis of literature is very poor (and sometimes of date) and many recent relevant references are missing.

  1. Buet et al ;
  2. Braun et al;
  3. Shimoda et al;
  4. Fellah et al;
  5. Hinoki et al ;

Comments:

In your introduction and discussion, you must add and discuss relevant references I proposed.

Line 30 : « high irradiation resistance” of SiC/SiC is only demonstrated for 3rd generation SiC fibers (not the case of the fibers used in this study) : You must add a reference and be rigorous on that point.

Line 57-63 :

  • You must add a chemical composition of the fibers ! That means excess of carbon relative to silicon content. Also, add a reference for the chemical composition values.
  • You must describe the processing of these composites, especially the PIP process: how many cycles, what is the maximal temperature during CVD and PIP process, porosity of composites, etc.. These data are important for discussion

Line 65-66 :

Description of tensile test is poor! You must describe the methodoly in details

Line 70-77:

You only talk about tensile strength but where are the experimental curves ? What is the modulus evolution with temperature ? Analysis based only on tensile strength values is difficult even impossible.

Figure 3 and line 80-87

The fracture morphologies of composites are interesting but for me there is no differences between 1200 and 1300°C and the morphology at 1500°C must be discuss in details. For that authors should have made the same magnification and orientation of pictures for 1500°C !

Line 90-97

The methodology for testing fibers must be described in details or a reference must be add by authors. More, what kind of test is used ? mono-filament tests ? tows tests ? what is the gauge length….

Table 2

Authors must add reference for values and must compare to the literature values. Are the values estimations? if yes authors should precise and/or take litterature values

Table 3

Same as table 2. Authors must give the references for all the values. There are some recent references for Pyrocarbon or matrix properties. The data are quite important regarding the very low values of radial residual stress obtained. A slight modification of these mechanical parameters can highly affect the calculation.

Authors only consider residual radial stress; they should consider all the residual stresses (longitudinal and circumferential) as in J. Braun et al reference for a rigorous discussion.

Figure 6 and line 165-173

The residual radial stress is very low for all conditions and I am not sure that the low differences between materials are significant. More, these residual stresses do not take account of densification of materials during high temperature tests (above processing temperature ?). In addition, these are calculations based on data of table 3, which are not measurements but sometimes estimations (especially for matrix and pyrocarbon properties). This point must be discussed relative to relevant references. Authors only take account of mechanical considerations but chemical interactions could have an influence.

Figure 8. and discussion

Same as figure 6, differences are low relative to uncertainties of the mechanical parameters of table 3 and what is the influence of densification of phases at 1300°C ? Authors must discuss this point and for that must detail the processing maximum temperatures and parameters.

Finally, your work is very interesting but you must discuss it with relevant references. You must also give details on data (references) and methodology you used.

Reviewer 2 Report

This is an interesting paper without any doubt. It reports of deals the effect of interface coatings on high temperature mechanical properties of SiC/SiC composite using domesitc Hi-Nicalon type SiC fibers. The authors described the investigated the tensile properties of SiC/SiC composites at high temperature using domestic Hi-Nicalon type SiC fiber, and illuminate the physical mechanism of mechanical property variations by combining theoretical analysis and experiments. The presented data are reliable and useful. However, the paper needs some improvement only after which it can be published.

  • In my opinion, the authors of the manuscript should necessarily refer to the newer publications from 2018-2020 in the "Introduction" section. The authors of the manuscript refer in their work to publications from 1995-2017. It seems to me that there are many curious and interesting works on microstructure and mechanical properties of SiC/SiC composite that have been published in recent years, which could be referred to in the "Introduction" part in manuscript.
  • In the part regarding "material and methods", there is no information on what equipment was used to conduct the tests of tensile test of SiC/SiC composites at high temperature in air. In the following part of the manuscript were conducted of microscopic observations of samples, but also there is no information what kind of microscope was used to observe.
  • The authors of the manuscript showed in their research that SiC fibers still retain high strength at 1300℃ and the strength retention rate is close to 80% on the based on research in situ on the Fiber Ultra-High Temperature Performance Testing System in Aerospace Research Institute of Materials & Processing Technology. Could the authors relate the obtained results to literature data? Do other studies also show that it indicates that the variation of fiber strength is not the key factor that causes the sharp reduction of tensile strength of SiC / SiC composites at high temperature?
  • Signs and symbols: Please, always use internationally accepted signs and symbols for units, SI units and give all dimensions according to the standard style of the Journal. Example: "GPa".
  • Please, give all Table headings according to the standards of Journal. A style guide is included at the “Instructions for Authors” menu on the website of the journal. Please systematized in Table 2 designation of subscripts to be identical to those which that are included in the test.
  • The above paper may be of possible technical interest. However, for submitting it to the Editorial review process English of the whole paper will mandatorily require considerable language revision efforts. Authors should take the help of some reliable example professional agency to proofread their manuscript or of a native English speaker with some technical knowledge of the subject.

Round 2

Reviewer 1 Report

You made some good modifications. Anyway, you must check and complete your references because some are not relevant and you must add some relevant references. There is a lot of recent articles dealing with the mechanical behavior of SiC/SiC composites. You should have much more references in your article.

Last, language revision is needed !

just a few comments

comments:

Line 32 : ref 3 is not representative for this purpose.
Line 34 : ref 5 and 6 are not dedicated to nuclear applications of SiC/SiC ! there are a lot of recent relevant references for that ! See Braun et al, in composites Part.A, 2019 for exemple that could be used also for residual thermal stress analysis.
Line 42 : there are 2 relevant articles of Buet: one is more recent (2014 in J. European. Ceram. soc.) and should be consider because it deals also with new thermal residual stresses calculations !
Line 50 : interface not interfase
Line 53-56 is to reconsider ! language revision is needed
line 59-60 : language revision is needed
Line 65 : Preform not perform
Line 67-68: the properties...are closed not property...is close
Line 71 : again preform not perform !
line 83 :language revision is needed
line 83-84: "The load direction is parallel to the longitudinal direction of the fiber" this does not mean anything.
Line 92 : "cureves" should be replace by curves
Line 93-94 : "It can be seen from the results that the 93 stress-strain curves at 800 ℃ is similar with that of RT" grammatical error ! if is similar it should be curve not curves

line 104-105 :Language revision is needed

Figure 3 : use the same scale for each column for a rigorous comparison analysis

Line 175 : ref 3 is not relevant for that ! this article deals with BN interface. Use relevant references (buet et al, 2014; braun et al, 2019...)
Line 178-184 ; authors don't take account of the densification process of PIP-SiC matrix during heating above 850°C.
table 4 : you must add references for these values in the description of table

Reviewer 3 Report

Line 2: Effect of Interface Coatings;  either Effect or Interface Coatings (grammary)

Line 4: Domesitc; Domestic

Line 7: why small letters for the name and capital ones?

Line 38: temperature[7]; an indent necessary

Line 50: interfase; interface

Line 55: have development; have made/reached/demonstrated development

Lines 69 and 71: performs; not preforms?

Line 73: are repeated…is less…; were…was…

Line 83: 2mm/min; 2 mm/min

Lines 79-85: suggest to use the same tense, and not either Past or Present

Lines 84-85: Five samples for each state are averaged; rather results?

Line 94: curves; curve

Line 106: characteristic; characteristics

Line 121: 5min; 5 min

Lines 121-122: rates of tensile testing was …; rate…

Lines 152-153: curve of fracture toughness  enhancement K  as a function of fiber/matrix interface strength can be obtained; relations of …

Line 168: that; those

Line 175 and 176: reference[3]; reference [3]

Line 178: with SiC fiber; as the SiC fiber

Line 178: 850nm; 850 nm

Line 215: temperature exceed; … exceeds

Line 226: reduction[9]; reduction [9]

Line 228: temperature[10-11]; temperaturę [10,11]

Line 229: bonding[21]; bonding [21]

Line 235: Reducing; reducing

Line 241: 100nm; 100 nm
